# Investigation on the In Situ Ti_2_AlC/TiAl Composite with a Homogenous Architecture by Adding Graphene Nanosheets

**DOI:** 10.3390/ma15165766

**Published:** 2022-08-21

**Authors:** Bo Hou, Aiqin Wang, Pei Liu, Jingpei Xie

**Affiliations:** 1College of Materials Science and Engineering, Henan University of Science and Technology, Luoyang 471023, China; 2Provincial and Ministerial Co-Construction of Collaborative Innovation Center for Non-Ferrous Metal New Materials and Advanced Processing Technology, Luoyang 471023, China

**Keywords:** Ti_2_AlC/TiAl composite, homogenous architecture, microstructure, mechanical properties

## Abstract

The Ti_2_AlC/TiAl composite with a homogenous architecture was fabricated via spark plasma sintering (SPS) using Ti/Al/GNSs composite powders, after ultrasonic mechanical stirring, as raw materials. The phases, microstructure, compressive properties and Vickers hardness of the composite were methodically characterized. We observed the transformation of graphene nanosheets from multi-layer to few-layer by the ultrasonic dispersion and the uniform distribution of few-layer graphene nanosheets in composite powders by ultrasonic mechanical stirring. The composite is mainly composed of rod-shaped Ti_2_AlC particles and a TiAl matrix, and the formation of rod-shaped morphology with the long axis along the (0001) plane is due to the fact that the growth rate of Ti_2_AlC parallel to the (0001) plane is much higher than the growth rate along the [0001] direction. The compressive stress and strain of the as-prepared Ti_2_AlC/TiAl composite reach 1451.2 MPa and 19.7%, respectively, which are better than some Ti_2_AlC/TiAl composites using graphite as the carbon source, and the Vickers hardness remains between 400~500 HV. The fracture morphologies show the deformation and fracture features of Ti_2_AlC particles, i.e., lamellae kinking and laminated tearing, which could increase the toughness of TiAl alloys.

## 1. Introduction

In recent decades, the extreme service environment has adopted increasingly stringent requirements for the performance of aerospace structural components due to the rapid development of aerospace and modern industries. TiAl alloys, as a new generation of high-temperature structural materials, have attracted much attention due to their remarkable physical and mechanical properties such as low density (3.7~4.1 g/cm^3^), high melting point, high specific strength and good creep resistance [1,2,3,4]. However, TiAl alloys exhibit poor ductility and formability at room temperature due to poor interfacial adhesion and compositional segregation, and these shortcomings greatly limit their wider practical applications [5,6]. To overcome the above problem, quite a few investigators have developed various methods to enhance the ductility of TiAl alloys at room temperature in recent years such as heat treatment [7,8,9], alloying [10,11,12], thermal–mechanical [13,14,15] and composite technology [16,17,18,19,20]. In the above-mentioned methods, the composite technology has engaged much attention because of its ability to compensate for the shortcomings of a single material and to take advantage of the “synergy” between the reinforcement and the matrix. Among the TiAl-based composites currently studied, TiB_2_, Al_2_O_3_, Ti_5_Si_3_ and Ti_2_AlC have been considered as suitable reinforcement for TiAl alloys. Notably, Ti_2_AlC is a ternary-layered compound that combines extraordinary properties of metal and ceramic due to the moderately strong Ti–C covalent bonds and weak Ti–Al metallic bonds such as high fracture resistance, excellent damage resistance, high stiffness and low thermal expansion. Moreover, the coefficients of thermal expansion of Ti_2_AlC (8.8 × 10^−6^ K^−1^) and TiAl (12 × 10^−6^ K^−1^) are also extremely close to each other; thus, it has been identified as the most outstanding reinforcement for TiAl alloys [21,22,23].

Compared with ex situ processes, the preparation of Ti_2_AlC/TiAl composites via in situ processes has become the dominating method for researchers because of its prominent advantages, such as small-size Ti_2_AlC particles, stable thermodynamic properties and clean interface [24,25,26,27]. At present, the reactive synthetic Ti_2_AlC/TiAl composites with a homogenous architecture are mostly based on the Ti-Al-C system with graphite as the carbon source. Both Song et al. [28] and Fang et al. [29] have prepared relatively uniform Ti_2_AlC/TiAl composites using graphite as the carbon source by vacuum arc melting, and found that the mechanical properties of composites were improved. Wei et al. [30] also synthesized relatively uniform Ti_2_AlC/TiAl composites by vacuum hot pressing using graphite as carbon source, and the results showed that the yield stress and hardness of composites were improved. Currently, carbon nanotubes (CNTs) and graphene with one-dimensional and two-dimensional structural features have attracted the widespread interest of researchers compared to graphite, a zero-dimensional carbon nanomaterial. The research results of Ti_2_AlC/TiAl composites with carbon nanotubes instead of graphite as the carbon source prepared by Zhu et al. [31] and Shrinivas et al. [32] show that carbon nanotubes could ensure the formation of fine grain structure and affect the reaction kinetics through promoting the formation of carbides. Graphene nanosheets (GNSs), as a typical representative among two-dimensional materials, possess more excellent properties than carbon nanotubes such as strength, thermal conductivity, electrical conductivity, toughness and stiffness. Therefore, the preparation of Ti_2_AlC/TiAl composites with a homogenous architecture using graphene nanosheets with scale advantage as the carbon source is expected to further improve mechanical properties [33,34,35]. Notably, the main prerequisite for achieving a uniform distribution of Ti_2_AlC-reinforcing particles is to ensure a uniform distribution of graphene nanosheets in the composite powders. However, to the authors’ knowledge, the preparation of Ti_2_AlC/TiAl composites with a homogenous architecture using Ti, Al and graphene nanosheets as a reaction system has not been reported so far. 

In this paper, we prepared the Ti_2_AlC/TiAl composite with a homogenous architecture via spark plasma sintering (SPS) using the Ti/Al/GNSs composite powders, after ultrasonic mechanical stirring, as raw materials, and the microstructure and compressive properties of the composite were further characterized. In addition, the growth mechanism of the rod-shaped Ti_2_AlC particles and the strengthening and toughness mechanism of the as-prepared Ti_2_AlC/TiAl composite were studied.

## 2. Experimental Methods

The schematic diagram of the fabrication process for Ti_2_AlC/TiAl composites with a homogenous architecture is shown in Figure 1 and could be divided into three steps: First of all, the original multi-layer graphene nanosheets were placed in anhydrous ethanol solution and dispersed into few-layer graphene nanosheets through ultrasonic dispersion. Secondly, based on the ingredient ratio of Ti-44Al-2.5C, the Ti/Al/GNSs composite powders are obtained by ultrasonic mechanical stirring using the few-layer graphene nanosheets, Ti powders (99.99% purity, <25 μm) and Al powders (99.99% purity, <25 μm). The operation process is as follows: The few-layer graphene nanosheets were obtained in anhydrous ethanol solution (0.5 g powders, 500 mL solution) by ultrasonic dispersion at a power of 840 W for 100 min, then we slowly added the Ti powders and Al powders, and introduced the mechanical stirring to continue sonication for 120 min with a stirring speed of 90 r/min. Afterwards, the composite powders were dried for 10 h at 70 °C in a vacuum oven. Thirdly, the composite powders were filled into the graphite mold with two graphite punches pressed at both ends, and then remained at 1623 K for 10 min by spark plasma sintering, with a heating rate of 100 °C/min and the press of 45 MPa to generate the Ti_2_AlC/TiAl composite. Eventually, the sample was machined for microstructure and mechanical properties testing.

The equipment used for GNSs dispersion was the ultrasonic cleaner (JP-060S), and the equipment used for stirring the composite powders was the constant-speed electric stirrer (JJ-1B). The phase composition of the as-prepared composite was detected by X-ray diffraction (XRD, Bruker D8 advanced) with Cu Ka radiation in a 2θ range of 25–85°. The polished sample surface was chemically etched in an etchant solution with volume fraction of 5% HNO_3_ + 5% HF + 90% H_2_O for about 20 s. The microstructure and element distribution were characterized by scanning electron microscopy (SEM, JSM-2100) and transmission electron microscopy (TEM, Talos F200X) with energy dispersive spectroscopy (EDS). The electronic universal testing machine (ZUAG-I250 KN) was used to carry out the compression test. The specimen size for the compression test was Φ5 mm × 10 mm and the constant loading rate was 0.5 mm/min. The Vickers hardness measurements were performed on the FV-810 tester (FUTURE-TECH, Qiutian, Japan) with a diamond indenter, loaded at 100 gf, 200 gf and 500 gf for 10 s. The sample for TEM observation was processed to 80 μm by mechanical thinning and then further machined to thin zones using the precise ion polishing system (PIPS, Gatan 691).

## 3. Experimental Results and Analysis

### 3.1. Morphology of Powders

Figure 2a,b shows the morphologies of the initial Ti powders and Al powders with a spherical structure. As indicated in the inset in Figure 2(c-1), the initial graphene nanosheets possess a multi-layer structure, and they show slight agglomeration due to the larger specific surface area. The morphology of graphene nanosheets after ultrasonic dispersion is shown in Figure 2c, and it can be seen that they transform into a dispersed and few-layer structure with the average particle size of 5–10 μm and thickness of 3–10 nm. It can be observed from Figure 2d that the few-layer graphene nanosheets uniformly distribute in the composite powders, which is very beneficial to the fabrication of Ti_2_AlC/TiAl composites with a homogenous architecture. The reason is that the high energy generated during the ultrasonic mechanical stirring process leads to the destruction of graphene nanosheets, and then most of powders will decrease in size and thickness, and a small portion may be transformed into separate graphene nanosheets.

### 3.2. Phase and Microstructure Characterization of the As-Prepared Ti_2_AlC/TiAl Composite

Figure 3 demonstrates the XRD pattern of the as-prepared composite, and it can be seen that the XRD pattern possess diffraction peaks of three phases, i.e., TiAl, Ti_3_Al and Ti_2_AlC. Furthermore, to observe the peak positions of three phases more clearly, the local enlarged images of 35–45° and 70–80° are shown in Figure 3(3-1) and Figure 3(3-2). The above illustration shows the typical crystal planes with high diffraction peak intensities in three phases: TiAl(111), Ti_3_Al(222) and Ti_2_AlC(0006). The result indicated that the chemical reaction designed in the experiment is relatively complete, and the Ti_2_AlC/TiAl composite is fabricated by using the reaction.

To determine the distribution of above phases, Figure 4 shows the SEM image and corresponding EDS analysis of the as-prepared Ti_2_AlC/TiAl composite. It can be clearly observed from Figure 4a that the white rod-shaped particles are uniformly distributed in the matrix, and the length and thickness are about 15 μm and 5 μm, respectively. The element distribution maps in Figure 4b indicated that these white rod-shaped particles are C-rich and Al-poor regions. In addition, the element distribution line in Figure 4c also shows that the intensity of Al and C element displays an obviously opposite trend when passing through the white rod-shaped particles, i.e., Al element decreases and C element increases. The point analysis result of the marked regions in Figure 4a is shown in Figure 4d. The atomic ratio of Ti: Al: C in the white rod-shaped particles is approximately equal to 2:1:1 and the atomic ratio of Ti: Al: C in the matrix area is close to 1:1:0. According to the XRD analysis results in Figure 3, we could conclude that these white rod-shaped particles are Ti_2_AlC and the matrix is TiAl, which indicates that the Ti_2_AlC/TiAl composite with a homogenous architecture had been fabricated with graphene nanosheets as the carbon source. In addition, the uniformity of the Ti_2_AlC particles in our composite was slightly better than that of the Ti_2_AlC/TiAl composites synthesized by Shu et al. [36] and Chen et al. [37] using graphite as carbon source, and their experimental results showed that Ti_2_AlC particles distribute in the grain boundaries with a cluster form. 

Apart from the micro-scale rod-shaped Ti_2_AlC particles under SEM, we also observed nano-scale rod-shaped Ti_2_AlC particles by transmission electron microscopy (TEM), as indicated in Figure 5. Figure 5a shows the TEM morphology of the rod-shaped Ti_2_AlC and TiAl matrix, and Figure 5b shows the EDS analysis result of marked area in Figure 5a, indicating that the atomic ratio of Ti and Al elements is close to 1:1. Figure 5c is the selected-area electron diffraction (SAED) pattern from the TiAl matrix. It could be determined from the calibration result that Figure 5c is the electronic diffraction pattern corresponding to the [$\overset{-}{1}$01] zone axis of TiAl. Figure 5d,e shows the high-resolution transmission electron microscopy (HRTEM) image and fast Fourier transform (FFT) pattern of the square area in Figure 5a, respectively. Figure 5e is the FFT pattern of Ti_2_AlC along the [11$\overset{-}{2}$
0] zone axis. The inverse fast Fourier transform (IFFT) of the square area in Figure 5d is exhibited in Figure 5f. As demonstrated in Figure 5f, the atomic stacking sequence of Ti_2_AlC can be regarded as the sequence of BABABAB along the [0001] direction, where the underlined letters correspond to Al layers and the non-underlined letters correspond to Ti layers, and the result is consistent with the Ti_2_AlC of the layered structure described previously [21,22,23].

Based on the location and morphology of the nano-scale rod-shaped Ti_2_AlC particles, we consider that they formed in the solid phase transition stage during the cooling process, and it has been discussed in our and others’ previous work [38,39]. A large amount of C will solid-dissolve in the TiAl matrix during the heating process, and then, when the temperature is lower than the α → α + γ temperature during the cooling process, the C whose solid solubility decreases with the decrease of the temperature will precipitate from the TiAl matrix in the form of Ti_2_AlC. Furthermore, we consider that the formation mechanism of the morphology of rod-shaped Ti_2_AlC can be explained by its crystal structure. The Ti_2_AlC possess a hexagonal layered structure: two Ti_6_C octahedra and an Al layer are alternately arranged along the [0001] direction. The Ti_2_AlC(0001) plane consists of the same kinds of atoms, while the Ti_2_AlC along the [0001] direction consists of different kinds of atoms. Due to the layered atomic arrangement, the growth rate of Ti_2_AlC parallel to the (0001) plane is higher than that along the [0001] direction. Therefore, Ti_2_AlC grew quickly and parallel to the (0001) plane when it precipitated from TiAl, resulting in a rod-shaped morphology with the long axis along Ti_2_AlC(0001) plane. In addition, due to the uniform dissolution of C atoms in the TiAl matrix, we consider that most of the nano-scale rod-shaped Ti_2_AlC particles may be uniformly distributed in the Ti_2_AlC/TiAl composite.

There is also a small amount of Ti_3_Al phase in the matrix besides the main phase of TiAl, and the reason for the formation of the Ti_3_Al phase is that the sintering temperature is higher than the eutectoid transformation temperature of the (TiAl + Ti_3_Al) two-phase. Figure 6a shows the TEM morphology of the Ti_2_AlC/Ti_3_Al interface, and the interface is clean and there are no other reactants. The EDS analysis result of marked area in Figure 6a is shown in Figure 6b, indicating that the contents of Ti and Al elements are quite different. Figure 6c shows the HRTEM image of the Ti_2_AlC/Ti_3_Al interface, and the corresponding FFT pattern of Ti_2_AlC and Ti_3_Al is inserted. Figure 6d shows the FFT pattern of the square area in Figure 6b, and Figure 6e is the indexed pattern of Figure 6c. The calibration result shows that the Ti_2_AlC is along the [11$\overset{-}{2}$0] zone axis and the Ti_3_Al is along the [$\overset{-}{1}$103] Ti_3_Al zone axis, and the (1$\overset{-}{1}$0$\overset{-}{1}$) plane of Ti_2_AlC and the (11$\overset{-}{2}$0) plane of Ti_3_Al are parallel to each other. Therefore, the following orientation relationship between Ti_2_AlC and Ti_3_Al results: $$[11\overset{-}{2}0]{\mathrm{Ti}}_{2}\mathrm{AlC}//\left[\overset{-}{1}103\right]{\mathrm{Ti}}_{3}\mathrm{Al},\;(1\overset{-}{1}0\overset{-}{1}){\mathrm{Ti}}_{2}\mathrm{AlC}//(11\overset{-}{2}0){\mathrm{Ti}}_{3}\mathrm{Al}$$

The IFFT image of the Ti_2_AlC/Ti_3_Al interface in Figure 6c is demonstrated in Figure 6f. As indicated in the Figure 6f, the blue circles and red circles represent the Ti_2_AlC atoms and Ti_3_Al atoms, respectively. It can be observed that the Ti_2_AlC(0001) plane is not parallel to the Ti_3_Al(2$\overset{-}{1}$$\overset{-}{1}$1) plane, which indicates that the interface between Ti_2_AlC and Ti_3_Al is an incoherent interphase boundary. 

### 3.3. Mechanical Properties of the As-Prepared Ti_2_AlC/TiAl Composite

Figure 7a exhibits the compression stress–strain curve of the as-prepared Ti_2_AlC/TiAl composite. It can be seen that the composite shows apparent features of plastic yielding during the compression process, and the compressive stress and compressive strain reach 1451.2 MPa and 19.7%, respectively. Figure 7b shows that the Vickers hardness of the as-prepared composite basically remains between 400~500 HV under different loads (100 gf, 200 gf and 500 gf). Figure 7c shows the comparison diagram of the compression properties between the Ti_2_AlC/TiAl composite in our work and other TiAl matrix composites [24,27,30,39,40,41,42]. It can be seen that the mechanical properties of the Ti_2_AlC/TiAl composite using graphene nanosheets as the carbon source is better than those with graphite as the carbon source. In addition, it also shows a certain improvement in plasticity compared with our previous composite with a laminated architecture [39], which is mainly due to the effect of uniformly distributed micro-scale and nano-scale Ti_2_AlC particles. The results indicate that the Ti_2_AlC/TiAl composites with a homogenous architecture prepared in this experiment have a relatively good strength, ductility and Vickers hardness. 

Figure 8a shows the fracture morphologies of the as-prepared Ti_2_AlC/TiAl composite, and it can be seen that the surface exhibits complex fracture modes. Figure 8b shows the crack deflection and particle pulling-out of the TiAl matrix in the composite. According to the zigzag propagation of cracks, it can be understood that the propagation path is prolonged due to the obstruction of surrounding Ti_2_AlC particles. In addition, Ti_2_AlC particles also exhibit certain deformation and fracture features under load, as shown in Figure 8b–d, i.e., lamellae kinking and laminated tearing. The lamellae kinking and laminated tearing of Ti_2_AlC particles have the effect of crack arrest, thereby improving the fracture toughness of the TiAl matrix.

## 4. Strengthening and Toughening Mechanism

The as-prepared Ti_2_AlC/TiAl composite with a homogenous architecture exhibited good strength and plasticity at a certain strain rate, which is inseparable from the presence of micro-scale and nano-scale rod-shaped Ti_2_AlC particles. The strengthening and toughening mechanism could be seen more intuitively through the schematic diagram of the model, as shown in Figure 9. As shown in Figure 9a, the load transfer effect between the TiAl matrix and the Ti_2_AlC particles under the external force allows the load to be transferred from the soft TiAl phase to the hard Ti_2_AlC phase, thereby increasing the strength of the material. Moreover, it can be observed from Figure 9b that the rod-shaped Ti_2_AlC particles could be pinned at the TiAl matrix grain boundaries to limit the growth of grains. It is generally considered that the finer the TiAl matrix grains, the greater the hindrance to dislocation, and the greater the contribution to the strength of the material. Figure 9c shows that the fine nano-scale Ti_2_AlC particles could hinder the sliding of dislocation lines, thereby forming dislocation loops around the Ti_2_AlC particles due to the bypassing mechanism. Furthermore, a large number of high-density dislocations are also formed around the Ti_2_AlC particles due to the hindering effect on the movement of dislocations. Both dislocation loops and high-density dislocations enhance the strength of the material. Figure 9d shows the ductile deformation behavior of Ti_2_AlC particles, i.e., lamellae kinking and laminated tearing, which could consume the energy of crack propagation without failure, thus improving the toughness of the material. 

## 5. Conclusions

In summary, the Ti_2_AlC/TiAl composite with a homogenous architecture has been fabricated by spark plasma sintering using Ti/Al/GNSs composite powders as raw materials after ultrasonic mechanical stirring. The following conclusions could be drawn from the characterization results of the phase composition, microstructure, compressive properties and Vickers hardness of the as-prepared composite: (1)The ultrasonic dispersion achieves the transformation of graphene nanosheets from multi-layer to few-layer, and the ultrasonic mechanical stirring ensured the uniform distribution of few-layer graphene nanosheets in composite powders;(2)The as-prepared composite is mainly composed of rod-shaped Ti_2_AlC particles with the lengths of about 15 μm and thicknesses of about 5 μm, as well as a TiAl matrix, and the formation of rod-shaped morphology with the long axis along the (0001) plane is due to the fact that the growth rate of Ti_2_AlC parallel to the (0001) plane is much higher than the growth rate along the [0001] direction;(3)The compressive stress and strain of the as-prepared Ti_2_AlC/TiAl composite reach 1451.2 MPa and 19.7%, respectively, and the Vickers hardness is about 400~500 HV, which maintain relatively good mechanical properties compared with some Ti_2_AlC/TiAl composites using graphite as carbon source;(4)The strengthening mechanism of the as-prepared Ti_2_AlC/TiAl composites is primarily due to the load transfer strengthening, refinement strengthening and Orowan strengthening, and the toughness mechanism is mainly attributed to the deformation and fracture of Ti_2_AlC particles, i.e., lamellae kinking and laminated tearing.

## Figures and Tables

**Figure 1 materials-15-05766-f001:**
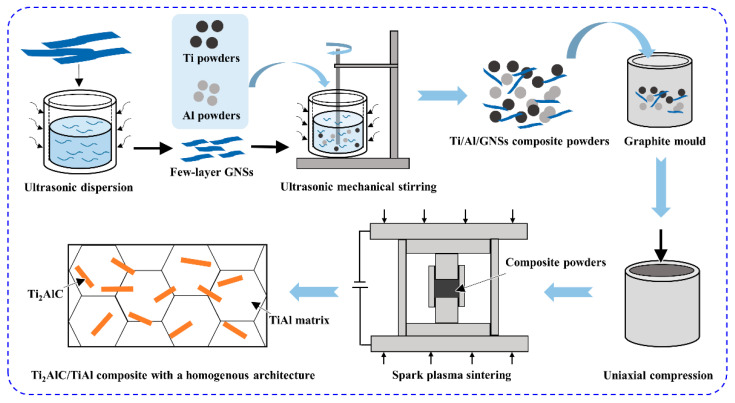
The schematic diagram of fabrication process for Ti_2_AlC/TiAl composite with a homogenous architecture.

**Figure 2 materials-15-05766-f002:**
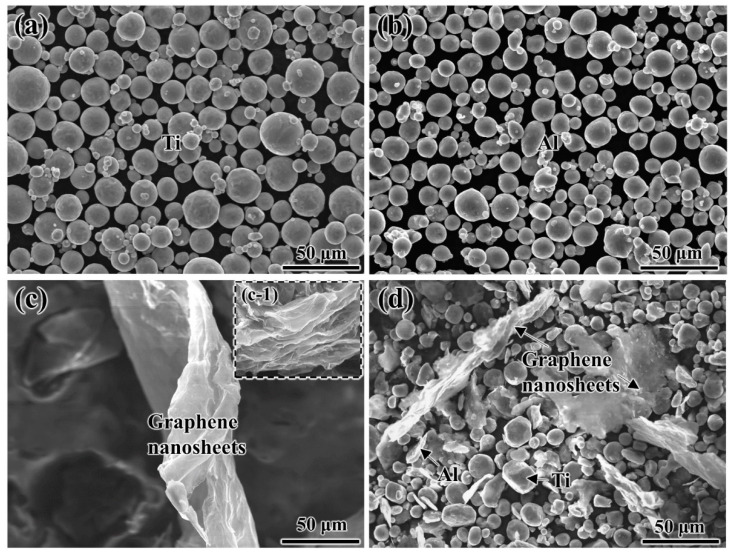
The SEM image of powders used for the fabrication of Ti_2_AlC/TiAl composite: (**a**) initial Ti powders; (**b**) initial Al powders; (**c**) few-layer graphene nanosheets; (**c-1**) initial graphene nanosheets; (**d**) Ti/Al/GNSs composite powders.

**Figure 3 materials-15-05766-f003:**
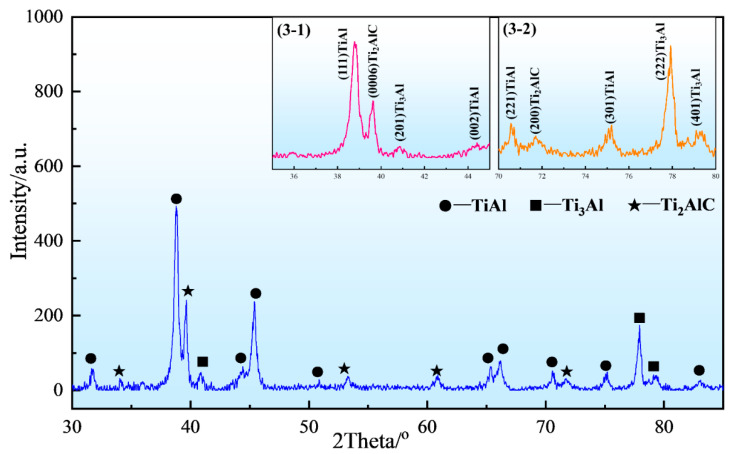
The XRD pattern of the as-prepared Ti_2_AlC/TiAl composite: (**3-1**) the local enlarged image of 35–45°; (**3-2**) the local enlarged image of 70–80°.

**Figure 4 materials-15-05766-f004:**
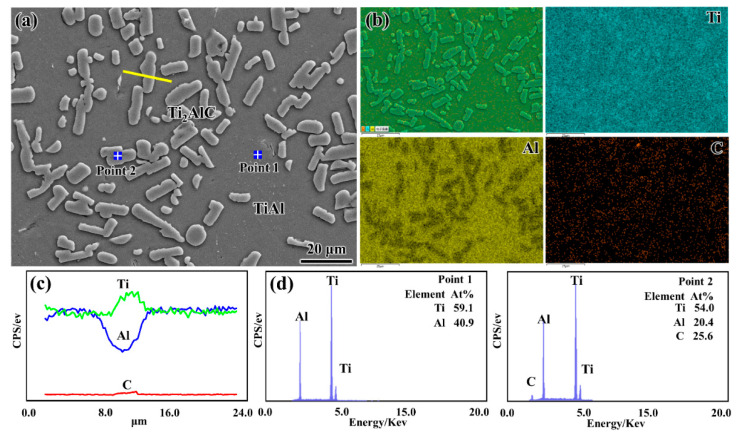
The SEM image and corresponding EDS analysis of the as-prepared Ti_2_AlC/TiAl composite: (**a**) SEM image; (**b**) element distribution maps in (**a**); (**c**) element distribution lines along the yellow line in (**a**); (**d**) point analysis of the regions marked in (**a**).

**Figure 5 materials-15-05766-f005:**
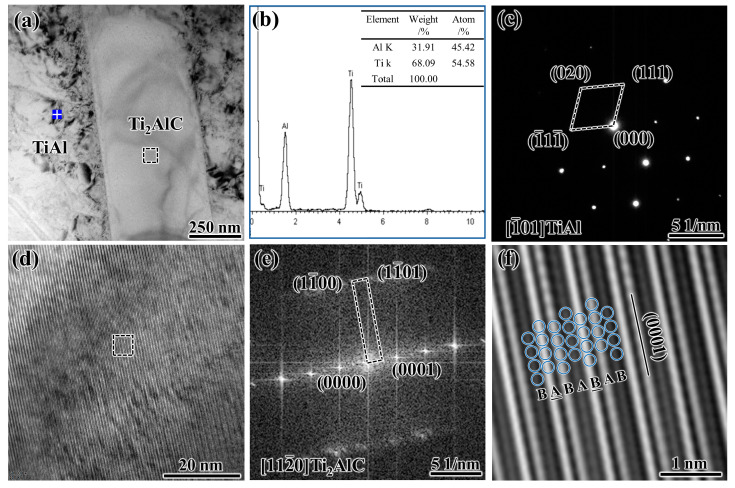
The TEM analysis of the rod-shaped Ti_2_AlC and TiAl in the as-prepared Ti_2_AlC/TiAl composite: (**a**) TEM image; (**b**) EDS analysis of the area marked in (**a**); (**c**) SAED pattern of the area marked in (**a**); (**d**) HRTEM image of Ti_2_AlC; (**e**) FFT pattern of Ti_2_AlC; (**f**) IFFT image of the square area in (**d**).

**Figure 6 materials-15-05766-f006:**
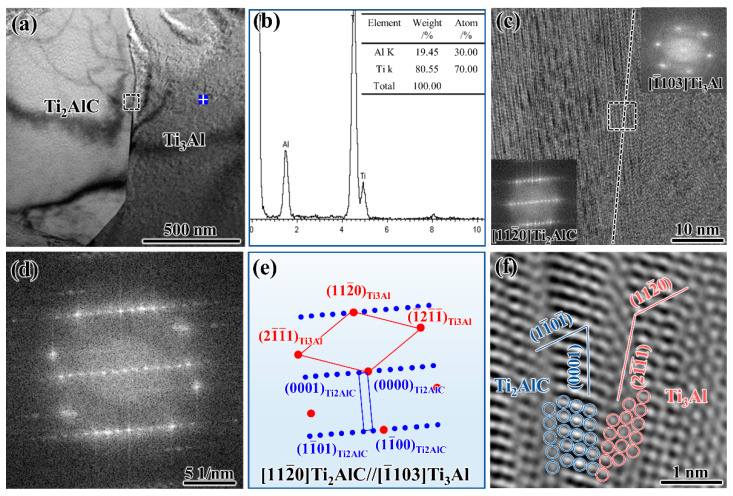
The TEM analysis of the Ti_2_AlC/Ti_3_Al interface: (**a**) TEM image; (**b**) EDS analysis of the area marked in (**a**); (**c**) HRTEM image of Ti_2_AlC/Ti_3_Al interface; (**d**) FFT pattern of Ti_2_AlC/Ti_3_Al interface; (**e**) Indexing of the FFT pattern in (**a**); (**f**) IFFT image of the square area in (**c**).

**Figure 7 materials-15-05766-f007:**
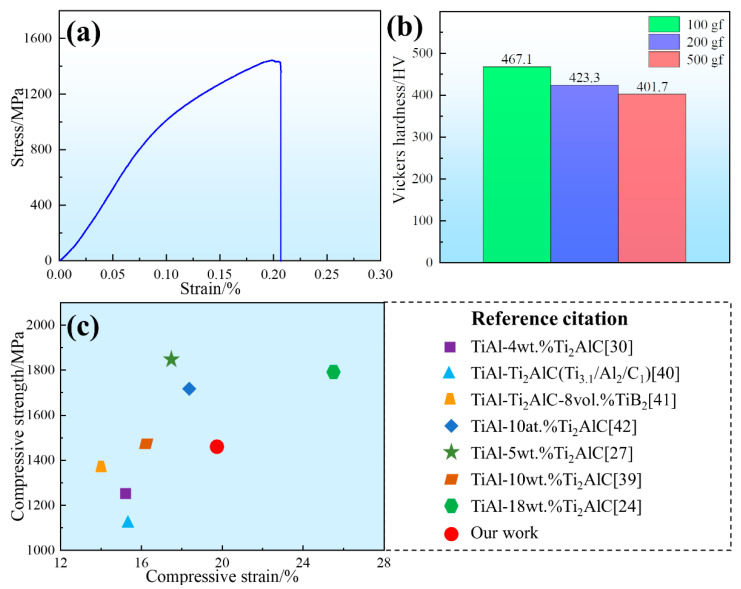
The compressive and hardness properties of the as-prepared Ti_2_AlC/TiAl composite: (**a**) compressive stress–strain curve; (**b**) Vickers hardness under loads of 100 gf, 200 gf and 500 gf; (**c**) comparison of the compressive properties of other Ti_2_AlC/TiAl composites in [24,27,30,39,40,41,42].

**Figure 8 materials-15-05766-f008:**
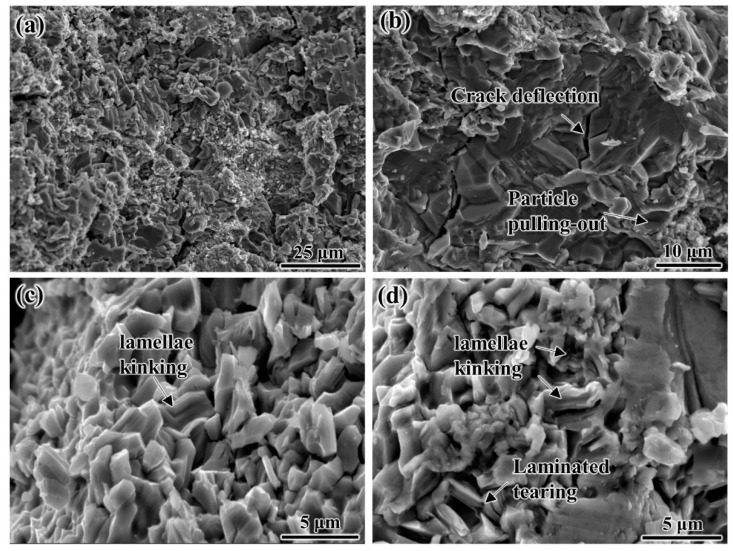
SEM images of fracture morphologies in as-prepared Ti_2_AlC/TiAl composite: (**a**) fracture morphology; (**b**) crack deflection and particle pulling-out; (**c**) lamellae kinking; (**d**) laminated tearing.

**Figure 9 materials-15-05766-f009:**
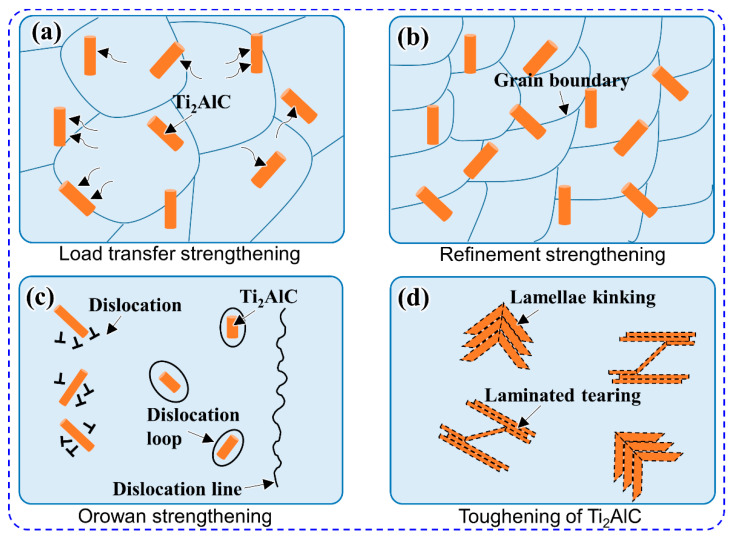
The schematic diagram for the strengthening and toughening mechanism of Ti_2_AlC/TiAl composite: (**a**) load transfer strengthening; (**b**) refinement strengthening; (**c**) Orowan strengthening; (**d**) toughening of Ti_2_AlC.

## Data Availability

All data generated or analysed during this study are included in this article.

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
