# Peer review of "Investigation on the In Situ Ti2AlC/TiAl Composite with a Homogenous Architecture by Adding Graphene Nanosheets"

_materials, 2022, doi:10.3390/ma15165766_

Round 1
Reviewer 1 Report
This paper describes a method to make a Ti2AlC/TiAl composite using spark plasma sintering with graphene nanosheets addition. Authors demonstrated mechanical strength and chemical composition study of the Ti2AlC/TiAl composite. The novelty of this paper is that authors utilized graphene nanosheets to improve the mechanical properties as the carbon source from graphene nanosheets can enhance the yield stress and hardness. In recent researches, a graphite was utilized as a carbon source to have Ti2AlC/TiAl composite which can have uniform dispersion and higher mechanic properties. Thus, in order to demonstrate the graphene adding method is the better method comparing to recent researches, authors need to show a comparison table in terms of mechanical performance and uniform dispersion. Additionally, it would be better to use ‘Graphene Nanosheets’ rather than using ‘GNSs’ in the title. Thus, I recommend that this paper is not suitable for publication in Materials at the current version.
Author Response
Thank you for your positive comments on our manuscript, we have revised the manuscript accordingly. Firstly, according to the literature [36] and [37], Shu et al. and Chen et al. prepared Ti2AlC/TiAl composites with graphite as the carbon source, and the microstructures showed some clustered Ti2AlC particles. Compared with the above results, the distribution of Ti2AlC particles in our composite is relatively uniform. We are very sorry that that we could not find a definite parameter to compare the uniformity of Ti2AlC particles, but we replaced the comparison table by adding a comparative description of uniform dispersion in the revised manuscript. Afterwards, the comparison of mechanical properties is shown in Fig. 7, and it could be seen that our composite have better strength or plasticity than other Ti2AlC/TiAl composites with graphite as the carbon source [27, 39, 40, 42, 43]. Finally, the title in this paper has been changed from GNSs to graphene nanosheets.
The content of the revised manuscript is as follows: “In addition, the uniformity of the Ti2AlC particles in our composite was slightly better than that of the Ti2AlC/TiAl composites synthesized by Shu et al. [36] and Chen et al. [37] using graphite as carbon source, and their experimental results showed that Ti2AlC particles distribute in the grain boundaries with the cluster from”, “Fig. 7(c) shows the comparison diagram of the compression properties between the Ti2AlC/TiAl composite in our work and other TiAl matrix composites [24,27,39-43]. It could be seen that the mechanical properties of the Ti2AlC/TiAl composite using graphene nanosheets as the carbon source is better than those of graphite as the carbon source [39,40]” and “Investigation on the in-situ Ti2AlC/TiAl composite with a homogenous architecture by adding graphene nanosheets”
Please see the attachment for more details.

Reviewer 2 Report
The article is written in an interesting way and contains important results on the obtainment of composites based on TiAl alloys; the introduction contains a detailed motivation for conducting this study. This is the third article by the authors on this topic, and it proposes a new alternative method for obtaining Ti2AlC/TiAl composites with a homogenous architecture. I believe that this article can be published after clarification of some details of the study.
1. In your earlier work (ref. 24 and Vacuum, 2022, 201, 111124) you suggest using flake powders to obtain micro-laminated Ti2AlC, which then allows you to synthesize Ti2AlC/TiAl composite with micro-nano laminated architecture. Why should graphene nanosheets be used in this work if it is not required to create a laminated structure in the composite? Why was it impossible to use dispersed carbon powder?
2. You use agglomerated graphene nanosheets a few microns thick and synthesize micro-scale rod-shaped Ti2AlC particles of about the same thickness. Why are nano-scale rod-shaped Ti2AlC particles formed? Does mixing cause partial destruction of graphene nanosheets or are there separate graphene nanosheets of nanoscale thickness? Are nano-scale rod-shaped Ti2AlC particles uniformly distributed in the composite or observed mainly near micro-scale rod-shaped Ti2AlC particles?
3. It would be very interesting if you make a comparison between Ti2AlC/TiAl composite with a homogenous architecture and micro-nano laminated architecture and show what advantages Ti2AlC/TiAl composite with a homogenous architecture can have.
Author Response
Please see the attachment for more detalis.

Round 2
Reviewer 1 Report
Now, I agree to publish the manuscript in Materials at the current version.